DATA RELEASE

# Genome assembly of the deep-sea coral *Lophelia pertusa*

Santiago Herrera[1,*] and Erik E. Cordes[2]

1 Department of Biological Sciences, Lehigh University, Bethlehem, PA, USA
2 Biology Department, Temple University, Philadelphia, PA, USA

## ABSTRACT

Like their shallow-water counterparts, cold-water corals create reefs that support highly diverse communities, and these structures are subject to numerous anthropogenic threats. Here, we present the genome assembly of *Lophelia pertusa* from the southeastern coast of the USA, the first one for a deep-sea scleractinian coral species. We generated PacBio continuous long reads data for an initial assembly and proximity ligation data for scaffolding. The assembly was annotated using evidence from transcripts, proteins, and *ab initio* gene model predictions. This assembly is comparable to high-quality reference genomes from shallow-water scleractinian corals. The assembly comprises 2,858 scaffolds (N50 1.6 Mbp) and has a size of 556.9 Mbp. Approximately 57% of the genome comprises repetitive elements and 34% of coding DNA. We predicted 41,089 genes, including 91.1% of complete metazoan orthologs. This assembly will facilitate investigations into the ecology of this species and the evolution of deep-sea corals.

**Subjects** Genetics and Genomics, Animal Genetics, Marine Biology

# DATA DESCRIPTION

## Context

Stony corals (Order Scleractinia) are foundational species in marine seafloor ecosystems worldwide. Due to their ecological importance, more than 40 whole genome assemblies of shallow-water scleractinian corals have been published to date [1–3]. Although most commonly associated with warm, shallow, tropical reefs, scleractinian stony corals are at least as diverse in cold water, particularly below the sunlit surface ocean, i.e., deeper than 50 meters below sea level (mbsl) [4]. However, no genome assemblies for deep-sea or cold-water scleractinian corals have been made available.

Cold-water coral reefs support highly diverse communities comprising faunal biomass that is orders of magnitude greater than that found in the surrounding seafloor [5–7]. In addition to this tightly-associated community, cold-water corals may also serve as important breeding, nursery, and feeding areas for a multitude of fishes and invertebrates [8, 9]. These communities rely on the transport of surface productivity to depth because of the lack of photosynthetic symbionts in the corals. Like their shallow-water counterparts, deep-sea corals are subject to ongoing anthropogenic threats, from ocean warming and acidification [10] to oil pollution [11]. Among deep-sea corals, *Lophelia pertusa* (Linnaeus, 1758), also known as *Desmophyllum pertusum* (NCBI:txid174260, marinespecies.org:taxname:135161) [12], is one of the most ecologically important species. *Lophelia pertusa* is a scleractinian coral that builds reef structures (Figure 1). This coral has a nearly-cosmopolitan distribution, spanning from approximately

**Submitted:** 31 December 2022

* Corresponding author. E-mail: santiago.herrera@lehigh.edu

Preprint submitted at https://doi.org/10.1101/2023.02.27.530183

**Figure 1.** *In situ* images of the coral *Lophelia pertusa* in the Atlantic USA southeast shelf. (a) *Lophelia* reef. (b) Close-up of *Lophelia* polyps. (c) Collection of the *Lophelia* sample sequenced in this study using the hydraulic arm of the remotely operated vehicle (ROV) Jason. (d) *Lophelia* sample being placed in ROV Jason's biobox. Images (a) and (b) courtesy of NOAA OER, Windows to the Deep 2019. Images (c) and (d) courtesy of the Deep SEARCH program and copyright Woods Hole Oceanographic Institution.

80 mbsl off the coast of Norway to over 1000 mbsl on the Mid-Atlantic Ridge. Although *L. pertusa* is arguably the best-studied deep-sea coral species, a high-quality reference genome assembly is still missing. This hinders our understanding of the biology of this coral species, its ecological functions, and its capacity to survive anthropogenic threats.

Here, we present the genome assembly of *Lophelia pertusa,* the first one for a deep-sea scleractinian coral species. Only one other genomic-level DNA sequence dataset was published for *D. pertusum.* Emblem and collaborators [13] produced 73 million SOLiD ligation sequencing reads and 1.2 million 454 pyrosequencing reads with average lengths of 46 bp and 580 bp, respectively. The Emblem dataset was useful for detecting mitochondrial single nucleotide polymorphisms but needed higher coverage and to be more cohesive to produce a useful genome assembly. Our study used PacBio continuous long reads (CLR) data for the initial assembly, followed by proximity ligation data for scaffolding and RNA-seq data for annotation. Our approach yielded a genome assembly of comparable quality to those obtained from shallow-water scleractinian corals [14–17].

## Methods

### Sample collection

Branches of *Lophelia pertusa* were obtained from the Savannah Banks site, off the southeastern coast of the continental USA, Atlantic Ocean (latitude 31.75420, longitude −79.19442, depth 515 mbsl), while aboard the NOAA Ship *Ronald Brown* (expedition RB1903) using ROV *Jason* (Dive 1130) on April 17, 2019 (BioSample accession SAMN31822850). The branches were collected using a hydraulic robotic arm and stored in an insulted biobox until they reached the surface (Figures 1c,d). Once onboard the ship, they were immersed in cold RNALater (Thermo Fisher), left to soak in the refrigerator (4 °C) for 24 hours, and



then frozen at −80 °C. Samples remained at that temperature until DNA was purified in the laboratory.

### DNA purification

Polyp tissue was scraped from the skeleton and digested in 2% cetyltrimethylammonium bromide (CTAB) buffer with 0.5% β-mercaptoethanol for 15 minutes at 68 °C. The DNA was purified through two rounds of phenol: chloroform: isoamyl alcohol (25:24:1) and one round of chloroform: isoamyl alcohol (24:1) mixing and partitioning through centrifugation at 10,000 rpm for 10 minutes. The DNA was precipitated out of the solution with 100% isopropanol. The resulting pellet was washed with 70% ethanol, then air-dried and resuspended in Qiagen G2 buffer. DNA concentration was quantified using a Qubit fluorometer (Invitrogen). The DNA was further purified using the Blood & Cell Culture DNA Midi Kit (Qiagen kit #13343) following the manufacturer's protocol after one hour of protease digestion. The average DNA fragment size was determined using pulsed-field gel electrophoresis (PFGE).

### DNA sequencing

A total of 19.3 Gbp contained in 2.07 million CLR were generated using a PacBio Sequel sequencer. For this, a 20 kb PacBio SMRTbell library was constructed using Blue Pippin Size selection. Long-insert chromosome conformation capture Chicago [18] and Hi-C [19] libraries (one each) were constructed and sequenced on an Illumina Hiseq X sequencer (paired-end,150bp), yielding 46.7 Gbp (156 million pairs) for the Chicago library and 72.6 Gbp (242 million pairs) for the Hi-C library.

### De novo genome assembly

The analytical pipeline to generate the *de novo* assembly of *Lophelia pertusa* is depicted in Figure 2. *De novo* genome assembly of PacBio data was performed using the assemblers flye (v2.9; RRID:SCR_017016) [20], wtdbg2 (v2.5, RRID:SCR_017225) [21], and FALCON (RRID:SCR_016089) [22], in combination with the polishing tools NextPolish v1.3.1 [23] and Arrow as implemented in the Pacific Biosciences GenomicConsensus package (https://github.com/PacificBiosciences/GenomicConsensus), and the haplotig and contig overlap removal program purge_dups (v.1.2.3, RRID:SCR_021173) [24]. First, we generated an assembly with flye using default parameters, followed by purging with purge_dups and polishing with NextPolish (assembly A). Using default parameters, we generated a second assembly with wtdbg2 and polished it with NextPolish (assembly B). A third assembly was generated using FALCON, followed by polishing with Arrow and purging with purge_dups (assembly C). Assemblies A and B were combined by aligning the flye assembly against the wtdbg2 assembly using MUMmer (v4.0; RRID:SCR_018171) [25], followed by merging with Quickmerge v0.3 [26] (-hco 5.0 -c 1.5 -l 248998 -ml 5000). The resulting assembly was polished with NextPolish (assembly D). Assembly D was aligned against assembly C using MUMmer and merged with Quickmerge. Finally, the assembly resulting from merging assemblies C and D was polished with NextPolish and purged with purge_dups (assembly E). Assemblies generated with other programs were not included because they had lower assembly contiguity or completeness (see the 'Data validation and quality control' section, Table 1).



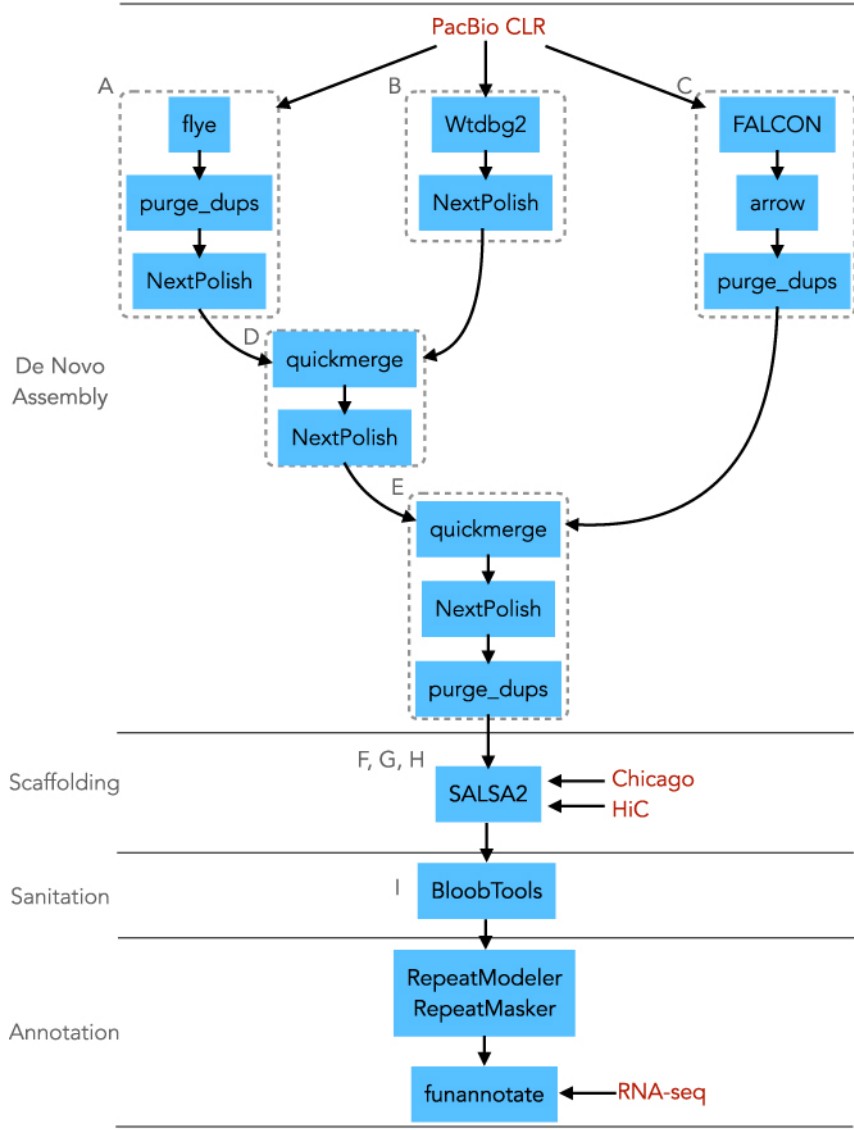

**Figure 2.** Flow chart depicting the assembly pipeline for the *Lophelia pertusa* genome. Dotted boxes indicate the different *de novo* assemblies. Letters indicate the designed nomenclature of each assembly as reflected in the text and Table 1. Data inputs are indicated in maroon font. Software packages are highlighted with blue background.

### Scaffolding

Assembly E was scaffolded with long-insert Chicago and Hi-C reads following the Arima Genomics mapping pipeline A160156 v02 (retrieved from https://github.com/ArimaGenomics/mapping_pipeline). First, the reads from the Chicago library were aligned to assembly E using the MEM algorithm of the program BWA (v0.7.17; RRID:SCR_010910) [27]. Chicago and Hi-C sequence data had mapping rates to the assembly of 96% and 98%, respectively, indicating high quality. Chimeric reads that mapped in the 3′ direction were excluded using the Arima-HiC Mapping pipeline filter_five_end.pl script [28]. Reads were combined into pairs with the two_read_bam_combiner.pl script and sorted using Samtools (v.1.10; RRID:SCR_002105) [29]. The program Picard tools (v2.26.6;



**Table 1.**  Statistics for *Lophelia pertusa* intermediate and final assemblies.

| Assembly ID | A | | | B | | C | | D | | E | | | F | G | H | I |
|---|---|---|---|---|---|---|---|---|---|---|---|---|---|---|---|---|
| Input data | PacBio CLR | PacBio CLR | PacBio CLR | PacBio CLR | PacBio CLR | PacBio CLR | PacBio CLR | PacBio CLR | PacBio CLR | PacBio CLR | PacBio CLR | PacBio CLR | PacBio CLR + Chicago | PacBio CLR + HiC | PacBio CLR + Chicago + HiC | H |
| Software | flye | flye + purge_dups | flye + purge_dups + NextPolish | wtdbg2 | wtdbg2 + NextPolish | FALCON + arrow | FALCON + arrow + purge_dups | quickmerge (A + B) | quickmerge (A + B) + Next polish | quickmerge (D + C) | quickmerge (D + C) + Next polish | quickmerge (D + C) + Next polish + purge_dups | SALSA2 (E + Chicago) | SALSA2 (E + HiC) | SALSA2 (F + HiC) | BlobToolKit |
| Sanitation | | | | | | | | | | | | | | | Prokaryot. | GC, cov., no-hit, undef |
| #contigs | 17,029 | 13,865 | 13,865 | 7,345 | 7,345 | 8,987 | 6,321 | 11,237 | 11,237 | 10,226 | 10,226 | 10,011 | 7,818 | 8,712 | 7,385 | **2,858** |
| #contigs (≥10 Kbp) | 11,441 | 8,278 | 8,514 | 5,789 | 5,863 | 8,710 | 6,044 | 6,218 | 6,248 | 5,431 | 5,438 | 5,223 | 3,284 | 4,073 | 2,910 | **2,019** |
| #contigs (≥25 Kbp) | 7,012 | 4,688 | 4,708 | 3,543 | 3,573 | 6,729 | 4,283 | 3,226 | 3,227 | 2,768 | 2,768 | 2,528 | 1,300 | 1,765 | 1,033 | **924** |
| #contigs (≥50 Kbp) | 4,449 | 3,271 | 3,274 | 2,119 | 2,121 | 4,292 | 2,835 | 2,142 | 2,145 | 1,843 | 1,844 | 1,630 | 897 | 1,148 | 676 | **652** |
| Largest contig (Kbp) | 1,284 | 1,284 | 1,278 | 2,222 | 2,198 | 1,100 | 1,100 | 3,039 | 3,036 | 3,134 | 3,136 | 3,136 | 5,013 | 6,202 | 10,677 | **10,677** |
| Total length (Kbp) | 781,392 | 615,714 | 618,620 | 546,887 | 548,050 | 685,805 | 487,642 | 620,046 | 619,797 | 635,659 | 635,608 | 588,242 | 589,370 | 588,927 | 589,296 | **556,859** |
| Total length (≥1 Kbp) | 781,186 | 615,508 | 618,417 | 546,886 | 548,049 | 685,805 | 487,642 | 619,842 | 619,593 | 635,455 | 635,406 | 588,040 | 589,174 | 588,731 | 589,104 | **556,857** |
| Total length (≥5 Kbp) | 774,389 | 608,711 | 611,986 | 545,841 | 547,109 | 685,800 | 487,637 | 613,412 | 613,222 | 629,083 | 629,053 | 581,686 | 583,014 | 582,517 | 583,001 | **556,159** |
| Total length (≥10 Kbp) | 751,665 | 585,996 | 590,000 | 536,436 | 537,966 | 683,319 | 485,155 | 594,011 | 593,904 | 611,231 | 611,203 | 563,836 | 566,568 | 565,396 | 566,844 | **551,248** |
| Total length (≥25 Kbp) | 680,889 | 529,945 | 530,627 | 498,949 | 499,636 | 648,954 | 455,537 | 547,800 | 547,280 | 570,231 | 570,093 | 522,303 | 537,194 | 530,454 | 539,228 | **534,434** |
| Total length (≥50 Kbp) | 589,893 | 480,070 | 480,059 | 449,330 | 448,950 | 685,805 | 403,091 | 509,550 | 509,117 | 537,749 | 537,651 | 490,800 | 523,329 | 509,117 | 526,899 | **525,028** |
| GC (%) | 39.57 | 39.59 | 39.57 | 39.22 | 39.40 | 39.36 | 39.37 | 39.55 | 39.55 | 39.54 | 39.54 | 39.55 | 39.55 | 39.55 | 39.55 | **39.53** |
| N50 (Kbp) | 114 | 138 | 137 | 249 | 248 | 123 | 142 | 331 | 329 | 455 | 452 | 467 | 901 | 824 | 1,440 | **1,614** |
| N75 (bp) | 51 | 59 | 58 | 84 | 82 | 65 | 69 | 82 | 82 | 117 | 117 | 109 | 413 | 219 | 553 | **689** |
| L50 | 1,817 | 1,229 | 1,243 | 586 | 590 | 1,582 | 977 | 455 | 457 | 366 | 366 | 329 | 186 | 155 | 94 | **83** |
| L75 | 4,373 | 2,935 | 2,976 | 1,509 | 1,527 | 3,487 | 2,200 | 1,453 | 1,458 | 1,038 | 1,040 | 953 | 420 | 495 | 258 | **219** |
| #N's/100 kbp | 2.23 | 3.01 | 0 | 0.00 | 0 | 0.00 | 0.51 | 0 | 0 | 0.05 | 0.0 | 0.5 | 191.90 | 116.74 | 238.68 | **248.99** |
| Busco (metazoa, *n* = 954) | | | 87.4 / 4.4 / 3.4 / 4.8 | | 90.7 / 0.9 / 2.8 / 5.6 | | 75.1 / 2.1 / 4.2 / 18.6 | | 87.9 / 4.7 / 2.1 / 5.3 | | | 88.6 / 2.4 / 2.1 / 6.9 | 89.0 / 2.1 / 2.1 / 6.8 | 88.9 / 2.1 / 2.1 / 6.9 | 89.0 / 2.1 / 2.1 / 6.8 | **88.9 / 2.2 / 2.1 / 6.8** |

RRID:SCR_006525) [30] was used to add read groups to the resulting bam file and remove PCR duplicates. The program SALSA2 v2.2 [31, 32] (-e GATC -m yes) was used for scaffolding assembly E with the mapped Chicago reads (assembly F). The Hi-C reads were mapped to assembly H using the same procedure described above and re-scaffolded with SALSA2 (assembly H).

### *Sanitation*

The program BloobToolKit v2.2 [33] was used to identify non-target scaffolds from assembly H. First, scaffolds were queried against the nucleotide collection database (nt) from the National Center for Biotechnology Information (NCBI), retrieved on May 5, 2020, using NCBI BLAST (RRID:SCR_004870) plus blastn (v2.10; RRID:SCR_001598) [34]. Scaffolds were then queried against the UniProt protein sequence database [35], retrieved on May 5, 2020, using DIAMOND blastx (v0.9.14.115; RRID:SCR_016071) [36]. Assembly coverage evenness was assessed by mapping the raw PacBio reads against assembly H using minimap2 (v2.24-r1122; RRID:SCR_018550) [37]. We excluded six scaffolds with significant matches to non-eukaryotic sequences (i.e., bacteria and viruses). We also excluded 4,531 scaffolds with significant deviations in coverage (<×0.01, >×65) or G.C. content (<26%, >52.5%) relative to the assembly-wide means (coverage = 3.27×, G.C. content = 39.81%) (assembly I). This Whole



Genome Shotgun (WGS) project was deposited at DDBJ/ENA/GenBank under the accession JAPMOT000000000.

### Annotation

Repetitive elements in the genome assembly I were identified *de novo* with the RepeatModeler v2.0.2 package (RRID:SCR_015027), including the programs RECON (v1.05; RRID:SCR_021170) [38] and RepeatScout (v1.06; RRID:SCR_014653) [39]. Repetitive elements were classified using RepeatClassifier v2.0.2 [40] and soft-masked using RepeatMasker (v4.1.2; RRID:SCR_012954) [40]. This procedure resulted in 57.37% of the genome assembly being masked.

The masked genome assembly was used for functional annotation using the Funannotate v1.8.9 pipeline [41]. First, we performed a *de novo* genome-guided transcriptome assembly using the Funnannotate *train* script with the *Lophelia pertusa* RNA-seq data published by Glazier and colleagues [42]. In short, (1) the RNA-seq data reads were normalized with Trinity (v2.8.5; RRID:SCR_013048) [43] and mapped to the masked genome assembly using HISAT2 (v2.2.1; RRID:SCR_015530) [44]; (2) a transcriptome assembly was generated with these mapped reads using Trinity; (3) the PASA (v2.4.1; RRID:SCR_014656) [45] program was used to produce a likely set of protein-coding genes based on transcript alignments.

Second, we performed gene prediction using the Funnannotate *predict* script (–repeats2evm –max_intronlen 30000 –busco_db metazoa). With this script, we (1) parsed transcript alignments to the genome to use as transcript evidence; (2) aligned the UniProtKB/SwissProt v2021_04 curated protein database [46] to the genomes and parsed alignments to use as protein evidence; (3) generated *ab initio* gene model predictions from the masked assembly with GeneMark-ES/E.T. (v4.68; RRID:SCR_011930) [47, 48], Augustus (v3.3.3; RRID:SCR_008417) [49], SNAP (v2013_11_29; RRID:SCR_007936) [50], and GlimmerHMM (v3.0.4; RRID:SCR_002654) [51], using PASA gene modes for training; (4) computed a weighted consensus of gene models from transcript, protein, and *ab initio* evidence using EVidenceModeler (v.1.1.1; RRID:SCR_014659) [52] (evidence source/weight: transcript/1, protein/1, Augustus/1, Augustus HiQ/2, GeneMark/1, GlimmerHMM/1, PASA/6, Snap/1); (5) filtered gene models to exclude transposable elements and lengths <50 aa; (6) predicted tRNAs using tRNAscan-SE (v2.0.9; RRID:SCR_010835) [53]. In total, 37,945 coding genes and 3,144 tRNA genes were predicted in the genome assembly. The average gene length was 4,972 bp. This analysis indicates that approximately 34% of the *Lophelia pertusa* genome is coding DNA.

The protein products of the predicted coding gene models were functionally annotated using the Funnannotate *annotate* script. The following annotations were added: (1) Protein family domains from PFAM (v35.0; RRID:SCR_004726) using HMMer (v3.3.2; RRID:SCR_005305) to find sequence homologs [54]; (2) Gene and product names from UniProt D.B. (v2021_04; RRID:SCR_002380) using DIAMOND blastp v2.0.13 [55] alignments; (3) Orthologous groups, gene, and product names from eggNog (v5.0; RRID:SCR_002456) [56] using eggNOG-mapper (v2.1.6; RRID:SCR_021165) [57]; (4) Protease annotation from MEROPS (v12.0; RRID:SCR_007777) [58] using DIAMOND blastp; (5) Metazonan single-copy orthologs from the OrthoDB (v10; RRID:SCR_011980) [59] using BUSCO (v5; RRID:SCR_015008) [60]; and (6) protein families and gene ontology (GO) terms from InterPro (v87; RRID:SCR_006695) using InterProScan (v5.53; RRID:SCR_005829) [61]. This procedure yielded 24,665 EggNog annotations, 24,471 InterPro annotations, 16,020 PFAM annotations, 16,646 GO terms, and 1,086 MEROPS annotations.



| | Genome Size (Mbp) | Scaffolds | N50 (Mbp) | GC% | Genes | BUSCO% (n=954) | | | |
|---|---|---|---|---|---|---|---|---|---|
| | | | | | | S | D | F | M |
| *Lophelia pertusa* | 556.9 | 2,858 | 1.6 | 39.4 | 41,089 | 89.0 | 2.1 | 2.1 | 6.8 |
| *Orbicella faveolata* | 485.5 | 1,933 | 1.2 | 28.6 | 30,178 | 84.8 | 0.5 | 8.5 | 6.2 |
| *Stylophora pistillata* | 400.1 | 5,688 | 0.5 | 34.5 | 28,912 | 91.6 | 0.9 | 4.4 | 3.1 |
| *Pocillopora damicornis* | 234.4 | 4,393 | 0.3 | 36.4 | 23,077 | 90.4 | 0.4 | 4.2 | 5.0 |
| *Porites lutea* | 552.0 | 2,975 | 0.7 | 35.7 | 31,126 | 93.5 | 1.8 | 2.1 | 2.6 |
| *Acropora millepora* | 475.4 | 854 | 19.8 | 39.1 | 42,775 | 86.7 | 5.5 | 3.1 | 4.7 |

0.03 substitutions/site

**Figure 3.** Quality metrics for the final *Lophelia pertusa* genome assembly (I), compared to other reference genome assemblies of scleractinian corals. BUSCO percentages indicate the proportion of the 954 metazoan orthologs that are complete and single-copy (S), complete and duplicated (D), fragmented (F), and missing (M). The phylogeny shown on the left is the best-scoring maximum likelihood tree inferred from single-copy orthologs. All branches had 100% bootstrap confidence.

## Quality control

The quality of each assembly was assessed using Quast (v5.0.2; RRID:SCR_001228) [62] and BUSCO v5 [60] (genome analysis with the metazoan lineage orthologs dataset OrthoDB v10 [59]). The steps described in the *de novo* assembly and scaffolding pipelines were implemented to maximize the contiguity, measured by the N50 statistic, and the completeness, measured by the percentage of single-copy metazoan orthologs present, in the assembly. The final assembly, assembly I, had an N50 of 1.61 Mbp, 5 to 10 times greater than the N50 of initial *de novo* assemblies without merging or scaffolding (assemblies A, B, and C). Similarly, assembly I had 89% complete single-copy metazoan orthologs of the 954 surveyed, which was between 7% and 18% more than initial *de novo* assemblies. Quality metrics for the final assembly (I) are shown in Figure 3. Quality metrics for all intermediate assemblies (A–H) are shown in Table 1.

The quality of genome assembly I is comparable to those obtained from shallow-water scleractinian corals. For comparison, we retrieved available genome assemblies of scleractinian corals with RefSeq (RRID:SCR_003496) annotations from the NCBI's Genome database. This genome set comprised assemblies for the species *Orbicella faveolata* [14], *Stylophora pistillata* [17], *Pocillopora damicornis* [15], and *Acropora millepora* [16]. We also retrieved the genome assembly of *Porites lutea* from reefgenomics.org. The quality of each of these assemblies was assessed using Quast and BUSCO as described above. The *Lophelia pertusa* assembly I has greater contiguity (N50) than most of the other scleractinian genomes in our comparison (0.3–1.2 Mbp), except for *A. millepora* (19.8 Mbp). The completeness of the *Lophelia pertusa* assembly I (91.1% complete metazoan orthologs, including single-copy and duplicated) is similar to the other scleractinian genomes (85.1–95.3%). The assembly size and the number of predicted genes of *Lophelia pertusa* (556.9 Mbp and 41,089 genes) are also similar, although larger than the other scleractinian genomes (234.4–552.0 Mbp and 20,267–31,834 genes). In our comparison, we used 242 single-copy orthologs present in all species to infer phylogenetic relationships among them. The amino-acid sequences of these orthologs were aligned using MAFFT (v7.453; RRID:SCR_011811) [63] and concatenated for each species (the final concatenated alignment contained 16,619 amino-acid sites). A species phylogeny was inferred in RAxML (v8.2.12; RRID:SCR_006086) [64] using the GAMMA model of rate heterogeneity. Branch support

values were estimated through 500 rapid bootstrap replicates. The resulting tree topology is congruent with the most recent phylogeny for the group [65].

### Re-use potential

The assembly of the *Lophelia pertusa* genome will facilitate numerous investigations into the ecology and evolution of this important species. This reference resource will enable population-genomic studies of this species within the US exclusive economic zone and comparative studies with populations throughout the Atlantic Ocean, Gulf of Mexico, Caribbean Sea, and Mediterranean Sea. This genome assembly will also be instrumental in resolving the taxonomic position of *Lophelia pertusa* as a monotypic genus instead of its proposed placement as a species, or set of species, within the genus *Desmophyllum*. This annotated genome assembly is the first one for a deep-sea scleractinian coral and thus will provide insights into the evolutionary history of deep-sea corals and the genomic adaptations to the deep-sea environment.

### DATA AVAILABILITY

The sequencing data and metadata supporting the results of this article are available at the US National Library of Medicine, on NCBI under the BioProject accession PRJNA903949, BioSample accession SAMN31822850, WGS accession JAPMOT000000000, and SRA accessions SRR22387542 (Hi-C reads), SRR22387543 (Chicago reads), and SRR22387544 (PacBio reads). The RNA-seq data is available under the BioProject accession PRJNA922177. A voucher of the *Lophelia pertusa* specimen sequenced in this study is available at the Smithsonian Institution National Museum of Natural History under the accession number USNM 1676648. The data is also available in the GigaDB repository [66].

### DECLARATIONS

### List of abbreviations

CLR, continuous long reads; mbsl, meters below sea level; GO, gene ontology; NCBI, National Center for Biotechnology Information; nt, nucleotide collection database; ROV, remotely operated vehicle; WGS, Whole Genome Shotgun.

### Ethics approval and consent to participate

Not applicable.

### Consent for publication

Approved for publication by the Bureau of Ocean Energy Management.

### Competing Interests

The authors declare that they have no competing interests.

### Funding

Sample collection was achieved through the Deep SEARCH project, funded by the Bureau of Ocean Energy Management (contract M17PC00009 to TDI Brooks International) and the NOAA Office of Ocean Exploration and Research (for ship time). Additional support came from the NOAA Deep-Sea Coral Research and Technology Program. Data generation was supported by an award from the Institute for Genomics and Evolutionary Medicine (iGEM)

of Temple University to EEC. The National Academies of Sciences, Engineering, and Medicine Gulf Research Program, Early-Career Fellowship 2000013668 to SH supported analysis and writing time.

## Authors' contributions

SH and EEC conceptualized the project. EEC acquired and managed the funding, collected the samples, and provided computational resources. SH and EEC generated the data. SH curated the data, performed analyses, generated visualizations, and wrote the original draft. SH and EEC reviewed and edited the manuscript.

## Acknowledgements

Alexis Weinnig assisted with the collection of samples. Amanda Glazier assisted with laboratory logistics and the composition of the proposal to iGEM. We thank Andrea Quattrini for the helpful discussions. Thanks to the science parties, captains, and crews of the expedition RB1903 aboard the NOAA Ship Ronald H. Brown.

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
