## [Reviewer Report]

Reviewer name and names of any other individual's who aided in reviewer Takeshi TakeuchiDo you understand and agree to our policy of having open and named reviews, and having your review included with the published papers. (If no, please inform the editor that you cannot review this manuscript.)YesIs the language of sufficient quality?YesPlease add additional comments on language quality to clarify if needed
Are all data available and do they match the descriptions in the paper? YesAdditional CommentsAre the data and metadata consistent with relevant minimum information or reporting standards? See GigaDB checklists for examples <a href="http://gigadb.org/site/guide" target="_blank">http://gigadb.org/site/guide</a>YesAdditional CommentsIs the data acquisition clear, complete and methodologically sound?YesAdditional CommentsIs there sufficient detail in the methods and data-processing steps to allow reproduction?YesAdditional CommentsIs there sufficient data validation and statistical analyses of data quality? NoAdditional CommentsScaffolding with the Chicago and Hi-C libraries did not significantly improve the assembly. In general, Hi-C scaffolding can produce a chromosome-scale assembly. I would suggest that the authors describe the quality of the Chicago and Hi-C sequence data. For example, the mapping rates of the Chicago/Hi-C reads to the assembly should be informative.Is the validation suitable for this type of data?YesAdditional CommentsIs there sufficient information for others to reuse this dataset or integrate it with other data?YesAdditional CommentsAny Additional Overall Comments to the AuthorRecommendationMinor Revision

---

## [Reviewer Report]

Reviewer name and names of any other individual's who aided in reviewer Yang ZhouDo you understand and agree to our policy of having open and named reviews, and having your review included with the published papers. (If no, please inform the editor that you cannot review this manuscript.)YesIs the language of sufficient quality?YesPlease add additional comments on language quality to clarify if needed
Are all data available and do they match the descriptions in the paper? YesAdditional CommentsAre the data and metadata consistent with relevant minimum information or reporting standards? See GigaDB checklists for examples <a href="http://gigadb.org/site/guide" target="_blank">http://gigadb.org/site/guide</a>YesAdditional CommentsIs the data acquisition clear, complete and methodologically sound?YesAdditional CommentsIs there sufficient detail in the methods and data-processing steps to allow reproduction?YesAdditional CommentsIs there sufficient data validation and statistical analyses of data quality? YesAdditional CommentsIs the validation suitable for this type of data?YesAdditional CommentsIs there sufficient information for others to reuse this dataset or integrate it with other data?YesAdditional CommentsAny Additional Overall Comments to the AuthorThis is a fascinating study on the assembly of the first deep-sea scleractinian coral, Lophelia pertusa. The manuscript is well-written and easy to follow. I have gone through your manuscript and would like you to address the following concerns/comments before publication.

Line 47: 1.2 454 pyrosequencing reads means 1.2Gb 454 pyrosequencing reads?
Line 51-52: Please add some references.
Line 72: As far as I know, the DNA extraction process of stony corals is affected by calcium carbonate skeletons. How did you deal with this problem during the DNA extraction process?
References: Please double-check the references for errors. Italics for species names, capitalization of journal titles, and so on.RecommendationMinor Revision